# Late Holocene glacier and climate fluctuations in the Mackenzie and Selwyn Mountain Ranges, Northwest Canada

Adam C. Hawkins[1], Brian Menounos[1,2], Brent M. Goehring[3], Gerald Osborn[4], Ben M. Pelto[5], Christopher M. Darvill[6], Joerg M. Schaefer[7]

[1]Department of Geography, Earth, and Environmental Science, University of Northern British Columbia, Prince George, V2M 5Z9, Canada
[2]Hakai Institute, Campbell River, V9W 2C7, Canada
[3]Los Alamos National Laboratory, Los Alamos, 87545, USA
[4]Department of Geoscience, University of Calgary, Calgary, T2N 1N4, Canada
[5]Department of Geography, University of British Columbia, Vancouver, V6T 1Z4, Canada
[6]Department of Geography, University of Manchester, Manchester M13 9PL, England
[7]Department of Earth and Environmental Sciences, Lamont-Doherty Earth Observatory, Columbia University, Palisades, 10964, USA

*Correspondence to:* Adam C. Hawkins (ahawkins@unbc.ca)

**Abstract.** Over the last century, northwestern Canada experienced some of the highest rates of tropospheric warming globally, which caused glaciers in the region to rapidly retreat. Our study seeks to extend the record of glacier fluctuations and assess climate drivers prior to the instrumental record in the Mackenzie and Selwyn Mountains of northwestern Canada. We collected 27 [10]Be surface exposure ages across nine cirque and valley glacier moraines to constrain the timing of their emplacement. Cirque and valley glaciers in this region reached their greatest Holocene extents in the latter half of the Little Ice Age (1600-1850 CE). Four erratics, 10-250 m distal from late Holocene moraines, yielded [10]Be exposure ages of 10.9-11.6 ka, demonstrating that by ca. 11 ka, alpine glaciers were no more extensive than during the last several hundred years. Estimated temperature change obtained through reconstruction of equilibrium line altitudes show that since ca. 1850 CE, mean annual temperatures rose 0.2-2.3 °C. We use our glacier chronology and the Open Global Glacier Model (OGGM) to estimate that since 1000 CE, glaciers in this region reached a maximum total volume of 34-38 km$^3$ between 1765-1855 CE and have lost nearly half their ice volume by 2019 CE. OGGM was unable to produce modeled glacier lengths that match the timing or magnitude of the maximum glacier extent indicated by the [10]Be chronology. However, when applied to the entire Mackenzie and Selwyn Mountain region, past-millennium OGGM simulations using the Max Planck Institute Earth System Model (MPI-ESM) and the Community Climate System Model 4 (CCSM4) yield late Holocene glacier volume change temporally consistent with our moraine and remote sensing record, while the Meteorological Research Institute Earth System Model 2 (MRI-ESM2) and the Model for Interdisciplinary Research on Climate (MIROC) fail to produce modeled glacier change consistent with our glacier chronology. Finally, OGGM forced by future climate projections under varying greenhouse gas emissions scenarios predict 85 to over 97% glacier volume loss by the end of the 21st century. The loss of glaciers from this region will have profound impacts to local ecosystems and communities that rely on meltwaters from glacierized catchments.

42
43

## 1 Introduction

Between 1990-2020 CE, northwestern Canada warmed by 1.1 °C above the 1961-1990 CE average (Muñoz-Sabater, 2019, 2021), which contributed to the loss of an estimated $0.429 \pm 0.232$ km$^3$ of ice in the Mackenzie and Selwyn Mountains of eastern Yukon and Northwest Territories between 2000 and 2020 CE (Figure 1; Hugonnet et al., 2021). Glaciers in this region are clearly responding to recent climate warming, but proxy evidence of past climate change is scarce (Tomkins et al., 2008; Dyke, 1990). Reconstructions of when and how glaciers responded to past climate change provide one method for estimating paleoclimatic conditions, while also placing the rate of modern glacier change into a geologic context.

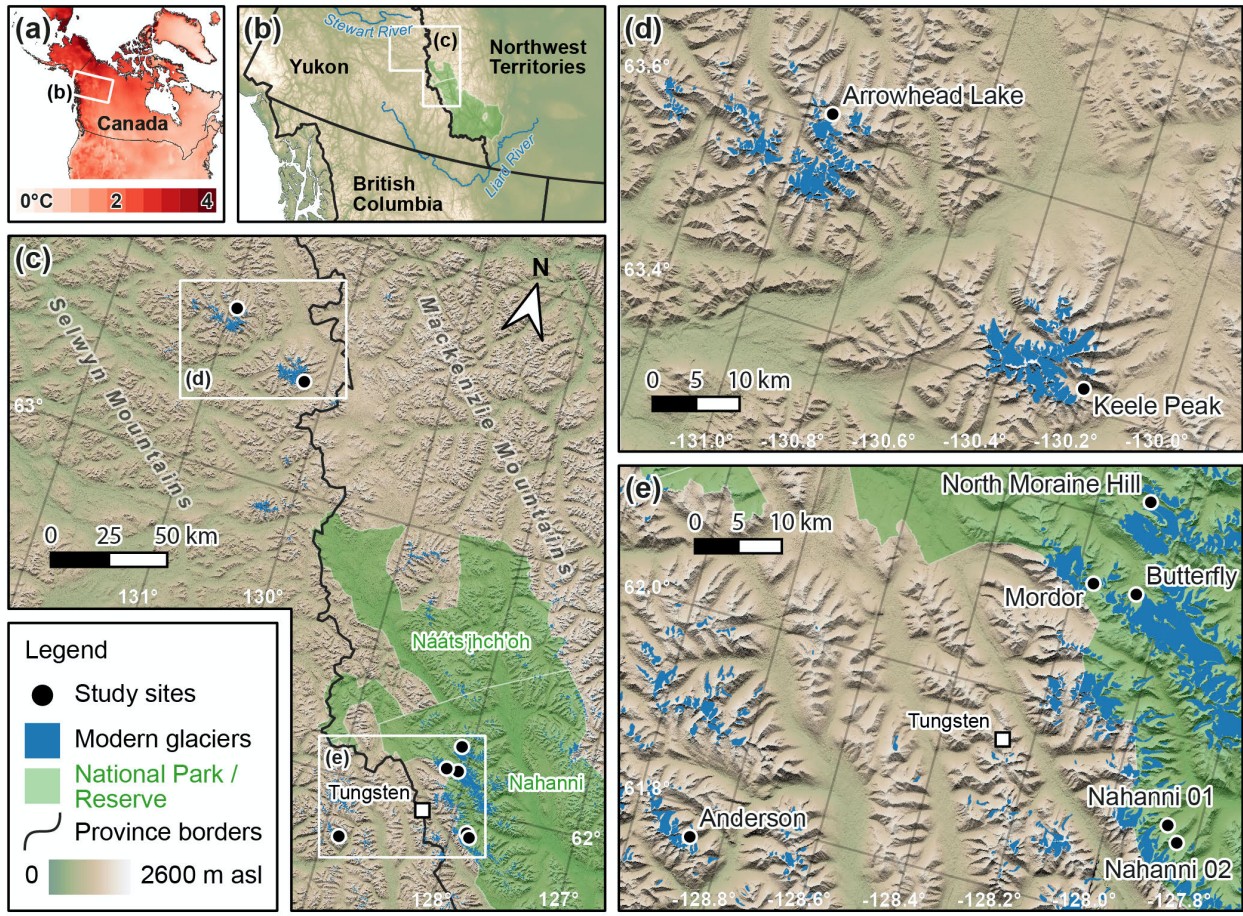

**Figure 1: Study area map of $^{10}$Be sampling locations.** Panel (a) is the temperature trend from ERA5land between 1950 and 2021 CE.

Few glacier change studies exist for the Mackenzie and Selwyn Mountains as compared to other mountainous regions in SW Yukon, British Columbia, and Alaska. Previous Quaternary research in this region focused on Pleistocene glacial deposits and Holocene rock glaciers (i.e. Duk-Rodkin

et al., 1996; Fritz et al., 2012; Menounos et al., 2017; Dyke, 1990). The remote location and related
logistical challenges of conducting fieldwork in this area are likely reasons this region is
underrepresented in Holocene climate reconstructions (e.g. Marcott et al., 2013).

The timing and magnitude of the most extensive Holocene glacier expansion in the eastern Yukon
and Northwest Territories, which places modern glacier retreat in context, remains uncertain.
Research in northern and interior Alaska indicates that glaciers reached their maximum Holocene
extents around 3.0-2.0 ka (Badding et al., 2013) while nearly all glaciers in southern Alaska and
western Canada reached their greatest Holocene positions around 1600-1850 CE, at the
culmination of the Little Ice Age (LIA, ~1300-1850 CE) (Menounos et al., 2009; Barclay et al.,
2009; Hawkins et al., 2021).

The primary objectives of our study are to develop a Holocene glacier chronology in the
Mackenzie and Selwyn mountains of eastern Yukon and Northwest Territories and use our glacier
chronology to estimate changes in climate responsible for these glacier fluctuations. We then
deepen our understanding of glacier activity in this area by estimating glacier volume change using
multiple models of past climate to force a glacier flowline model. Finally, we briefly evaluate
future glacier change in this region in response to various greenhouse gas emissions scenarios.
**2 Study area**
The Mackenzie and Selwyn ranges extend over 600 km from north of the Liard River in
northwestern British Columbia to the Stewart River and northern extent of the Mackenzie Range
in northern Yukon (Fig. 1). This region is covered by 650 km$^2$ of ice from nearly 1200 glaciers
situated among peaks that rise as high as 2952 m above sea level (Pfeffer et al., 2014). Bedrock
consists of faulted and folded Paleozoic sedimentary rocks with Early Cretaceous granitic
intrusions (Pfeffer et al., 2014; Cecile and Abbott, 1989). A portion of our study area is situated in
the Nahanni (Nááts'ihch'oh) National Park Reserve, which was expanded in 2009 to >30,000 km$^2$
(Demuth et al., 2014). Glacier runoff within the Nahanni National Park Reserve flows into the
Liard River watershed which later joins the Mackenzie River, eventually draining north to the
Beaufort Sea. Two of our nine field sites are located nearly 200 kilometers to the northwest of
Nahanni National Park Reserve and are situated on or adjacent to the Keele Peak massif, which is
similarly composed of Early Cretaceous granitic rock. Meltwater from our study sites on and near
the Keele Peak massif flows into the Stewart River, which flows west to the Yukon River and
eventually to the Bearing Sea. The watersheds in our study area are culturally and ecologically
important for the numerous First Nations communities who have lived on this land for millennia,
including the Dënéndeh, Kaska Dena, and Na-Cho Nyak Dun First Nations, among others.
**3 Methods**
Our glacier chronology originates from digitized glacier margins of aerial photos and satellite
imagery and constraining the age of late Holocene moraines using cosmogenic $^{10}$Be surface
exposure dating. Cosmogenic surface exposure dating relies on the accumulation of rare isotopes,
in this case $^{10}$Be, in the bedrock surface during periods of exposure at or near the surface of the
Earth (Gosse and Phillips, 2001). We use this chronology to estimate paleoclimate conditions in
the late Holocene using several methods. First, we estimate past and present equilibrium line
altitudes (ELA) using the maximum elevation of lateral moraines (MELM, LIA maximum only),
toe-to-headwall altitude ratio (THAR), and accumulation area ratio (AAR) and infer changes in
temperature and precipitation from estimated ELA changes (Braithwaite and Raper, 2009; Meier
and Post, 1962; Ohmura and Boettcher, 2018). We then estimate the temperature decrease needed
to grow glaciers to their late Holocene positions using a flowline glacier model. Additionally, we
perturb monthly temperature and precipitation from several General Circulation Model (GCM)
simulations of climate since 1000 CE to produce modeled glacier extents that most closely match
the terrestrial and remotely sensed record (Taylor et al., 2012) before evaluating past modelled
glacier volume change for all glaciers in the Mackenzie and Selwyn mountains. Finally, we model
future glacier change in this region under various Representative Concentration Pathways (RCPs;
Moss et al., 2010).
**3.1 Field site selection**
We selected sampling locations within the Mackenzie and Selwyn Mountain ranges using satellite
imagery, aerial photos, and digital elevation data to identify purported late Holocene moraines.
We consulted bedrock geologic maps of the area to locate sites that likely contained quartz-bearing
lithologies suitable for $^{10}$Be surface exposure dating (hereafter $^{10}$Be dating), which was then
confirmed in hand-samples in the field (Cecile and Abbott, 1989; Gordey, 1992). Helicopters and
floatplanes during late summer in 2014, 2016, and 2017 provided access to the field sites.
**3.2 Mapping of former and present glacier extents**
We manually digitized past glacier outlines for six of the nine glaciers sampled for [10]Be dating.
Those glaciers represent sites with multiple dated moraine boulders and morphologies better suited
for glacier flowline modeling. It is the author's understanding that only two of the glaciers included
in this study, North Moraine Hill and Butterfly glaciers, have formal names. The remaining
glaciers are referred to with informal names below. The resulting glaciers used in paleoclimate
reconstructions are Anderson, Mordor, North Moraine Hill, Butterfly, Keele Peak, and Arrowhead
glaciers (Fig. 2). We used imagery from airphotos between 1949 and the mid-1970's CE and
satellite imagery from 1985 CE, onward (SM Table 2). Air photos represent digitally scanned
negatives housed at the Canadian National Airphoto Library (NAPL). We georeferenced each
airphoto by manually selecting 40-60 ground control points (GCPs) on the air photographs and
high-resolution satellite imagery (e.g. large boulders, peaks, and ridges). We subsequently
performed a thin-plate spline transformation in GIS software (QGIS), visually inspecting the
georeferenced image for any obvious distortions. Portions of glacier outlines further from GCPs
have positional errors smaller than 20 m.

We used Landsat 5, 7, and 8 satellite imagery to delineate glacier margins at roughly 5-10 year
intervals from the mid-1980's onward (SM Figure 12). To aid in the manual digitization, we made
false color composites for each Landsat scene to highlight the glacier surface relative to non-
glaciated terrain. The surfaces of most glacier termini are debris free, which facilitated glacier
mapping. We mapped late Holocene glacier margins using high resolution satellite imagery from
Mapbox and PlanetLabs to delineate glacier trimlines and moraine crests. In areas with cloud cover
or snow-covered terrain, we used hillshades from ArcticDEM to help identify moraine ridges
(Porter et al., 2018).

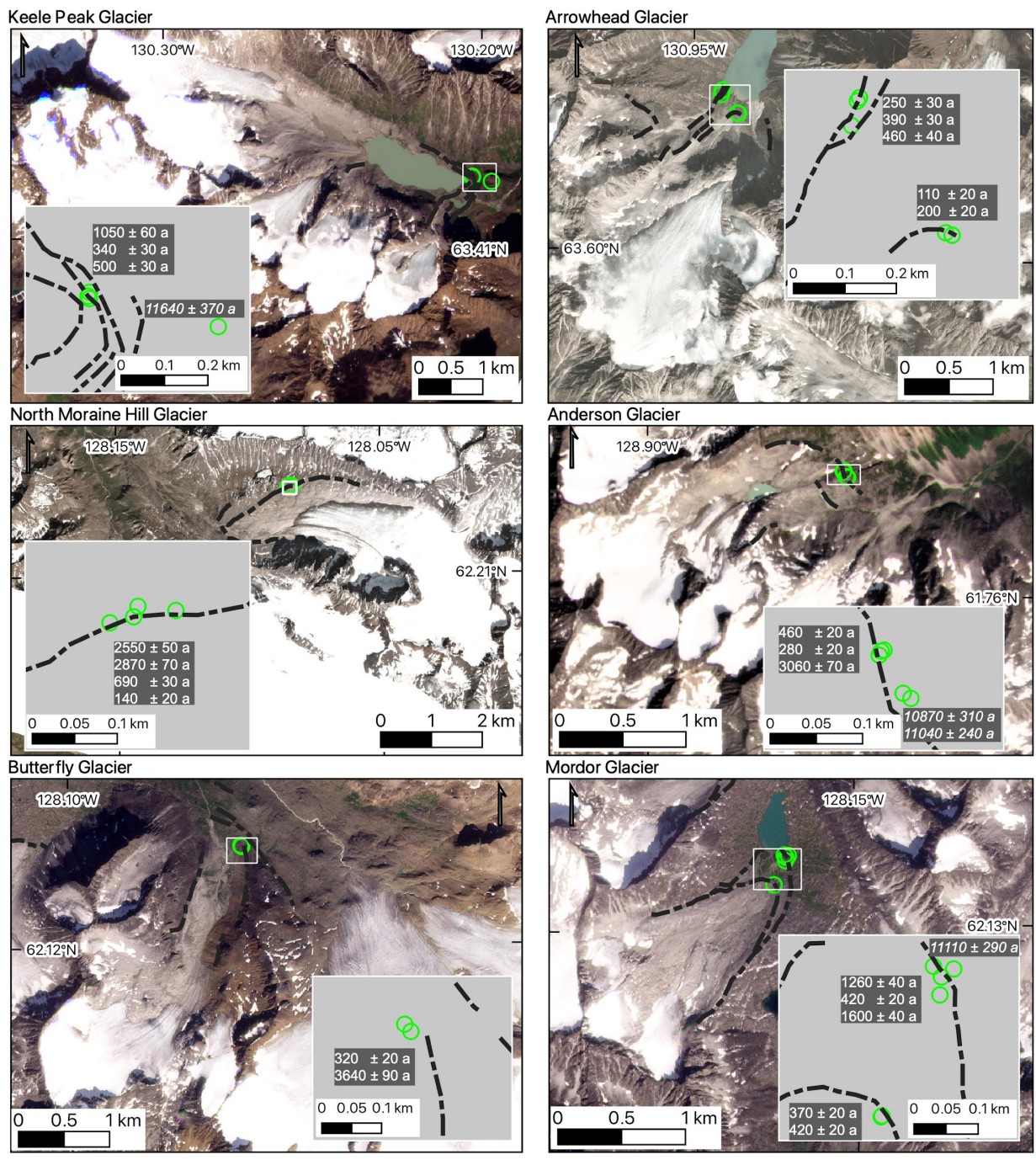


**Figure 2: Glaciers from which [10]Be samples were collected.** Sample locations are shown with green circles. Moraine crests are
depicted as black dashed lines. Exposure ages ± analytical errors for individual boulders are in text boxes, with erratic boulders
ages shown in italics. Grey insets show sampling sites at larger scale. Imagery is from PlanetLabs, acquired between July and
August, 2021 and 2022.

### 3.3 [10]Be field sampling

We targeted samples from large (generally taller than 1 m), granitic boulders on or near moraine crests (Fig. 2, SM Data). It is commonly assumed that large boulders on moraine crests are windswept such that snow cover is minimal, and their large size limits the chance of being previously covered by moraine material or moving following deposition (Heyman et al., 2016). Recent work by Tomkins and others (2021) provides evidence that sampling from the crests of moraines may not reduce the chance of geomorphic exposure age scatter, however at the time of sampling in this study, we followed the common practice of targeting boulders on moraine crests. Several erratic boulders directly overlying bedrock and distal to the moraine crests were sampled as well (SM Data). We measured topographic shielding of the incoming cosmic ray flux and boulder self-shielding using a Brunton compass and inclinometer, and then determined the location and elevation of each sample with a handheld GPS receiver with barometric altimeter. Samples were collected from the top surfaces of boulders using a concrete saw and hammer and chisel to collect approximately 1 kg of rock.

### 3.4 [10]Be laboratory procedures and AMS measurements

The Lamont-Doherty Earth Observatory Cosmogenic Nuclide Laboratory processed samples collected in 2014, and we analyzed the remaining samples in the Tulane University Cosmogenic Nuclide Laboratory. All samples were crushed, milled, and sieved to 250-750 μm. Physical and chemical isolation of quartz was completed following the procedures of Nichols and Goehring (2019). We isolated Be using standard chemical isolation procedures, including anion and cation exchange columns (Ditchburn and Whitehead, 1994; Schaefer et al., 2009). We included a process blank with every batch of ~eight samples (SM Table 3). We sent sample aliquots of extracted Be to either the Purdue Rare Isotope Measurement (PRIME) Laboratory or the Lawrence-Livermore National Laboratory (LLNL_CAMS) for AMS measurements, which were normalized to the standard KNSTD dilution series (Nishiizumi et al., 2007).

We calculated the exposure ages for all samples using version 3 of the online exposure age calculator formerly known as CRONUS-Earth, hosted by the University of Washington (https://hess.ess.washington.edu/). We used the default [10]Be reference production rates from the

"primary" calibration dataset (Borchers et al., 2016) and report individual sample ages using the
Lifton-Sato-Dunai (LSDn) scaling scheme and 1-sigma analytical errors (Table 1). No corrections
for burial by snow or surface erosion are applied to the moraines as snow depth and its variation
and rates of surface erosion are poorly constrained. We do, however, provide estimates of how
exposure ages may be influenced by snow cover (SM Table 4). Moraine ages are reported as the
median exposure age ± interquartile range to avoid the issue of using statistics that assume an
underlying distribution of the ages of the moraine boulders, a key requirement of parametric
approaches to characterize central tendency and dispersion (Menounos et al., 2017; Darvill et al.,

186    2022).

| Sample | Latitude | Longitude | Elevation (m asl) | Thickness (cm) | Shielding | Quartz (g) | Carrier added (g)[a] | 10Be/9Be ratio | 1 sigma uncertainty | Blank-corrected 10Be conc. (atoms/g)[b] | Blank-corrected 10Be conc. uncertainty (atoms/g) | Exposure age a (LSDn)[c,d] | Exposure age uncertainty | AMS Facility |
|---|---|---|---|---|---|---|---|---|---|---|---|---|---|---|
| **Nahanni Nat'l Park area** | | | | | | | | | | | | | | |
| *Nahanni 01* | | | | | | | | | | | | | | |
| 14NA-01 | 61.9075 | -127.8688 | 1500 | 1.64 | 0.934 | 15.01 | 0.183 | 6.20E-15 | 5.32E-16 | 5.18E+03 | 4.45E+02 | 300 | 30 | LLNL-CAMS |
| 14NA-02 | 61.9075 | -127.8686 | 1500 | 2.02 | 0.9272 | 15.068 | 0.1836 | 7.57E-15 | 5.09E-16 | 6.36E+03 | 4.27E+02 | 370 | 30 | LLNL-CAMS |
| 14NA-03 | 61.9079 | -127.8697 | 1515 | 1.9 | 0.9212 | 14.023 | 0.1834 | 7.79E-14 | 1.75E-15 | 6.97E+04 | 1.53E+03 | 4060 | 90 | LLNL-CAMS |
| | | | | | | | | | | | Median ± IQR | 370 ± 940 | | |
| *Nahanni 02* | | | | | | | | | | | | | | |
| 14NA-04 | 61.8924 | -127.8406 | 1550 | 1.6 | 0.9614 | 15.003 | 0.1828 | 1.36E-14 | 8.85E-16 | 1.14E+04 | 7.39E+02 | 410 | 40 | LLNL-CAMS |
| 14NA-06 | 61.8925 | -127.8404 | 1550 | 2.22 | 0.9595 | 15.005 | 0.1827 | 1.52E-14 | 1.33E-15 | 1.27E+04 | 1.10E+03 | 670 | 60 | LLNL-CAMS |
| | | | | | | | | | | | Median ± IQR | 640 ± 20 | | |
| *Butterfly Glacier* | | | | | | | | | | | | | | |
| 14NA-07 | 62.1299 | -128.0637 | 1710 | 2.12 | 0.9837 | 13.729 | 0.1833 | 7.02E-15 | 4.72E-16 | 6.46E+03 | 4.34E+02 | 320 | 20 | LLNL-CAMS |
| 14NA-09 | 62.1298 | -128.0635 | 1715 | 1.93 | 0.9824 | 15.032 | 0.1833 | 8.88E-14 | 2.10E-15 | 7.41E+04 | 1.78E+03 | 3640 | 90 | LLNL-CAMS |
| | | | | | | | | | | | Median ± IQR | 1980 ± 830 | | |
| *"Anderson" Glacier* | | | | | | | | | | | | | | |
| 16-AND-02 | 61.769 | -128.8705 | 1606 | 2.5 | 0.9656 | 27.207 | 0.2673 | 1.41E-14 | 4.76E-16 | 8.51E+03 | 3.46E+02 | 460 | 20 | LLNL-CAMS |
| 16-AND-03 | 61.769 | -128.8706 | 1607 | 2.5 | 0.9653 | 28.24 | 0.2676 | 9.20E-15 | 3.88E-16 | 5.09E+03 | 2.76E+02 | 280 | 20 | LLNL-CAMS |
| 16-AND-04 | 61.769 | -128.8706 | 1608 | 2.5 | 0.9656 | 39.269 | 0.2681 | 1.23E-13 | 2.29E-15 | 5.56E+04 | 1.19E+03 | 3060 | 70 | LLNL-CAMS |
| 16-AND-05 | 61.7686 | -128.87 | 1605 | 2.5 | 0.9628 | 50.011 | 0.2672 | 5.50E-13 | 1.46E-14 | 1.96E+05 | 5.57E+03 | *10870* | 310 | LLNL-CAMS |
| 16-AND-06 | 61.7686 | -128.8701 | 1606 | 2.5 | 0.9676 | 50.053 | 0.2674 | 5.64E-13 | 1.05E-14 | 2.00E+05 | 4.26E+03 | *11040* | 240 | LLNL-CAMS |
| | | | | | | | | | | | Median ± IQR | 460 ± 700 | | |
| *"Mordor" Glacier outer moraine* | | | | | | | | | | | | | | |
| 16-MOR-13 | 62.1301 | -128.1604 | 1765 | 2.5 | 0.9762 | 37.115 | 0.2567 | 6.13E-14 | 1.54E-15 | 2.74E+04 | 8.21E+02 | 1260 | 40 | LLNL-CAMS |
| 16-MOR-14 | 62.1302 | -128.1606 | 1764 | 2.5 | 0.9765 | 50.011 | 0.258 | 3.06E-14 | 8.47E-16 | 8.83E+03 | 4.96E+02 | 420 | 20 | LLNL-CAMS |
| 16-MOR-15 | 62.1298 | -128.1604 | 1765 | 2.5 | 0.9792 | 50.012 | 0.2583 | 1.02E-13 | 1.92E-15 | 3.45E+04 | 8.67E+02 | 1600 | 40 | LLNL-CAMS |
| 16-MOR-16 | 62.1302 | -128.16 | 1761 | 2.5 | 0.9769 | 50.016 | 0.2569 | 6.53E-13 | 1.53E-14 | 2.31E+05 | 5.95E+03 | *11110* | 290 | LLNL-CAMS |
| | | | | | | | | | | | Median ± IQR | 1260 ± 300 | | |
| *"Mordor" Glacier inner moraine* | | | | | | | | | | | | | | |
| 16-MOR-11 | 62.1281 | -128.1622 | 1785 | 2.5 | 0.9754 | 46.672 | 0.2567 | 2.51E-14 | 9.34E-16 | 7.98E+03 | 4.02E+02 | 370 | 20 | LLNL-CAMS |
| 16-MOR-12 | 62.1281 | -128.1622 | 1762 | 2.5 | 0.9754 | 50.023 | 0.2572 | 2.89E-14 | 8.32E-16 | 8.81E+03 | 3.47E+02 | 420 | 20 | LLNL-CAMS |
| | | | | | | | | | | | Median ± IQR | 390 ± 10 | | |
| *North Moraine Hill Glacier* | | | | | | | | | | | | | | |
| 16-MH-16 | 62.2256 | -128.0849 | 1870 | 2.5 | 0.9864 | 50.007 | 0.2569 | 1.67E-13 | 3.12E-15 | 5.93E+04 | 1.27E+03 | 2550 | 50 | LLNL-CAMS |
| 16-MH-17 | 62.2256 | -128.0844 | 1870 | 2.5 | 0.9861 | 50.002 | 0.2579 | 1.91E-13 | 3.65E-15 | 6.63E+04 | 1.52E+03 | 2870 | 70 | LLNL-CAMS |
| 16-MH-18 | 62.2257 | -128.0835 | 1869 | 2.5 | 0.986 | 50.005 | 0.2583 | 5.28E-14 | 1.49E-15 | 1.68E+04 | 6.81E+02 | 690 | 30 | LLNL-CAMS |
| 16-MH-19 | 62.2257 | -128.0834 | 1866 | 2.5 | 0.9855 | 50.022 | 0.2593 | 1.42E-14 | 7.06E-16 | 2.98E+03 | 4.57E+02 | 140 | 20 | LLNL-CAMS |
| | | | | | | | | | | | Median ± IQR | 1620 ± 1040 | | |
| **Keele Peak area** | | | | | | | | | | | | | | |
| *Keele Peak Glacier* | | | | | | | | | | | | | | |
| 17-KP-01 | 63.4201 | -130.2021 | 1548 | 2.5 | 0.9726 | 50.004 | 0.2587 | 5.47E-14 | 2.74E-15 | 1.94E+04 | 1.01E+03 | 1050 | 60 | PRIME |
| 17-KP-02 | 63.42 | -130.2021 | 1542 | 2.5 | 0.9726 | 49.995 | 0.2588 | 1.73E-14 | 1.27E-15 | 5.99E+03 | 4.68E+02 | 340 | 30 | PRIME |
| 17-KP-03 | 63.42 | -130.2021 | 1541 | 2.5 | 0.9694 | 48.52 | 0.259 | 2.47E-14 | 1.35E-15 | 8.91E+03 | 5.14E+02 | 500 | 30 | PRIME |
| 17-KP-04 | 63.4195 | -130.1961 | 1602 | 2.5 | 0.9869 | | 0.2587 | 4.50E-13 | 9.30E-15 | 2.16E+05 | 4.96E+03 | *11640* | 270 | PRIME |
| | | | | | | | | | | | Median ± IQR | 500 ± 180 | | |
| *Arrowhead Glacier outer moraine* | | | | | | | | | | | | | | |
| 17-AH-05 | 63.6162 | -130.9434 | 1410 | 2.5 | 0.9364 | 40.254 | 0.2593 | 9.10E-15 | 9.36E-16 | 3.76E+03 | 4.32E+02 | 250 | 30 | PRIME |
| 17-AH-06 | 63.6166 | -130.9432 | 1408 | 2.5 | 0.9364 | 44.016 | 0.2584 | 1.54E-14 | 1.18E-15 | 6.02E+03 | 4.94E+02 | 390 | 30 | PRIME |
| 17-AH-07 | 63.6166 | -130.9431 | 1413 | 2.5 | 0.9364 | 30.637 | 0.2593 | 1.27E-14 | 1.04E-15 | 7.08E+03 | 6.27E+02 | 460 | 40 | PRIME |
| | | | | | | | | | | | Median ± IQR | 390 ± 50 | | |
| *Arrowhead Glacier inner moraine* | | | | | | | | | | | | | | |
| 17-AH-08 | 63.6143 | -130.9396 | 1440 | 2.5 | 0.9517 | 50 | 0.2595 | 4.90E-15 | 7.40E-16 | 1.51E+03 | 2.79E+02 | 110 | 20 | PRIME |
| 17-AH-09 | 63.6143 | -130.9393 | 1440 | 2.5 | 0.9517 | 47.738 | 0.2594 | 8.66E-15 | 8.22E-16 | 3.01E+03 | 3.23E+02 | 200 | 20 | PRIME |
| | | | | | | | | | | | Median ± IQR | 150 ± 20 | | |

[a] Be Carrier for samples 14-NA* was 1038.3 ug/g, except samples 14-NA(02&07), whose carrier was 1038.8 ug/g. All remaining samples used a PRIME Be carrier with concentration of 1040 ppm.
[b] Isotopic ratios were meaured at either the Lawrence Livermore National Laboratory - Center for Accelerator Mass Spectrometry (LLNL-CAMS) or the Purdue Rare Isotope Measurement Laboratory (PRIME). Be-10/Be-9 ratios are not corrected for Be-10 detected in procedural blanks.
[c] Ages are calculated using version 3 of the online exposure age calculator formerly known as the CRONUS-Earth online exposure age calculator found at https://hess.ess.washington.edu/ (wrapper 3.0.2, muons: 1A, constants as of: 2020-08-26). All ages are calculated using the Lifton-Sato-Dunai "LSDn" scaling and the default production rate. Ages and errors are rounded to the nearest decade.
[d] The median exposure age and interquartile range (IQR) excludes the exposure age of erratics, whose ages are listed in italics.

**Table 1: 10Be sample information for all boulders sampled in this study.**

### 3.5 ELA reconstructions

Variations in the equilibrium line altitude of a glacier relate to long term changes in climate. Such variations have been used to estimate changes in either temperature or precipitation (Dahl and Nesje, 1992; Moore et al., 2022; Oien et al., 2022). Commonly used methods to reconstruct past ELAs include the maximum elevation of lateral moraines, toe-to-headwall altitude ratio, and

accumulation area ratio, among others. Each method offers advantages and limitations in
reconstructing past ELAs (Benn et al., 2005; Nesje, 1992; Porter, 2001; Osmaston, 2005). We use
the MELM, THAR, and AAR methods of ELA reconstruction to estimate glacier ELAs between
the Little Ice Age (ca. 1300-1850 CE) and modern time (2000-2021 CE).
To record the MELM for each glacier, we used high resolution satellite imagery and elevation data
from ASTER GDEM version 3 (NASA/METI/AIST/Japan Spacesystems and U.S./Japan ASTER
Science Team, 2019) to identify the highest elevation of preserved lateral moraines.
The THAR method assumes a glacier's ELA is positioned at a fixed ratio between the maximum
and minimum elevation of the glacier, shown in Eq. (1):
$ELA = minimum\ glacier\ elevation + (glacier\ elevation\ range \times THAR)$     (1)
Work by Meirding (1982) and Murray & Locke (1989) found that ratios of 0.35 to 0.4 yielded
satisfactory estimates of alpine glacier ELAs. Here, we use the mean ELA from a THAR of 0.35
and 0.4.
The accumulation area ratio assumes a fixed ratio of the accumulation area to the total area of a
glacier in equilibrium (Braithwaite and Raper, 2009; Meier and Post, 1962). Here, we assume the
AAR for glaciers in this region to be 0.6, which is generally considered to be the ratio of steady
state cirque and valley glaciers in NW North America (Porter, 1975).
We generated LIA and modern glacier hypsometries by clipping the ASTER DEM to the digitized
glacier extents. In this case, the modern glacier extents are from the latest satellite imagery used
for each glacier (imagery from 2017-2021 CE). We acknowledge that the modern DEM does not
account for the paleo surface of the glacier during the LIA and may negatively bias the paleo-ELA
(Porter, 2001).
For each ELA reconstruction method, we inferred the change in average temperature (dT) from
the Little Ice Age to present as a function of changing ELA by assuming an environmental lapse
rate of -6.5 °C km$^{-1}$.

The ELA of a glacier is also influenced by changes in precipitation. Ohmura et al. (2018; 1992)
empirically derive an equation (Eq. 2) to estimate the annual precipitation, $P$, in millimeters water
equivalent (mm w.e.) at the ELA of a glacier, given a mean summer (JJA) temperature $T$:
$$P = a + bT + cT^2,\tag{2}$$
where, $a = 966$, $b = 230$, and $c = 5.87$. We estimated changes in precipitation at the ELA of each
study glacier by assuming a modern (1986-2015 CE mean) JJA temperature ($T$) at the modern
ELA from the fifth generation European Centre for Medium-Range Weather Forecasts (ECMWF)
global climate atmospheric reanalysis (ERA5). We use our dT estimate from our ELA
reconstructions to yield Eq. 3:
$$P_{LIA} = a + b(T - dT) + c(T - dT)^2\tag{3}$$
We selected ERA5 2 m surface temperatures (Hersbach et al., 2020) from the grid cell nearest to
the study glacier and used the same -6.5 °C km$^{-1}$ lapse rate to approximate $T$ at the modern ELA.
**3.6 Glacier modeling**
**3.6.1 Open Global Glacier Model**
Our final method of ELA reconstruction uses the Open Global Glacier Model (OGGM; Maussion
et al., 2019) which is a modular, open-source model framework with the capacity to model glacier
evolution for all glaciers on Earth. The glacier model within OGGM is a depth-integrated flowline
model that solves the continuity equation for ice using the shallow ice approximation (Cuffey and
Paterson, 2010). Multiple flowlines for each glacier are calculated using a DEM clipped around
the glacier polygon using the routing algorithm of Kienholz et al. (2014). The default mass-balance
model used in OGGM begins with gridded monthly climate data, here the Climatic Research Unit
gridded Time Series (CRU TS) version 4.04 (Harris et al., 2020). The climate data feeds a
temperature index model described in Marzeion et al. (2012), incorporating a temperature
sensitivity parameter that is calibrated using nearby glaciers with observations of specific mass
balance (Zemp et al., 2021). Ice thickness is estimated by assuming a given glacier bed shape
(parabolic, rectangular, or mixed) and applying a mass-conservation approach that employs the
shallow-ice approximation. OGGM assumes that the "modern" glacier outline, sourced from the
Randolph Glacier Inventory (RGI), is from the same date as the DEM. Users are also able to supply
their own glacier outlines. More information on OGGM can be found on OGGM.org, or in
publications on the model (Maussion et al., 2019; Eis et al., 2021).
**3.6.2 Equilibrium run**
In our first experiment using OGGM, we started with the RGI polygons for the six of our study
glaciers targeted for surface exposure dating (Anderson, Mordor, Butterfly, North Moraine Hill,
Keele Peak, and Arrowhead glaciers). We then ran a 1000-year simulation under a constant
climate, iteratively adjusting a temperature bias relative to the average CRU TS climate centered
around 2000 CE (close to the RGI polygon date of most glaciers in the region) until the modeled
glacier reached equilibrium at or very near the glacier length indicated by the moraine record.
From these equilibrium run experiments, we produce three different estimates of ELA and
temperature change. First, the temperature lowering required to expand a glacier to its LIA length
was interpreted as the approximate temperature change from the LIA to 2000 CE. Second, we then
extracted the hypsometry of the modeled glacier at t=0 (modern extent) and t=1000 (LIA extent)
and estimated the modeled ELA using the same AAR method as described in section 3.5, again
assuming an AAR of 0.6. We can again apply the -6.5 °C km$^{-1}$ lapse rate to estimate the apparent
temperature change from modelled glacier extents between the two time periods. Third, for the
modern glacier extent, we extracted the elevation at which the modeled surface mass balance of
each glacier is equal to zero without any temperature bias. This represents the modern *climatic*
ELA and is not based on glacier morphology.
**3.6.3 Transient run**
In our next experiment with OGGM, we simulate changes in glacier volume in the Mackenzie and
Selwyn mountains using our glacier chronology to tune the climate model input. We used OGGM
to simulate the response of our five glaciers driven by monthly temperature and precipitation
variability from four Coupled Model Intercomparison Project Phase 5 (CMIP5) GCM runs
(CCSM4, MIROC-ESM, MPI-ESM-P, and MRI-ESM2; Taylor et al., 2012). All GCMs
incorporate volcanic, total solar irradiance, summer insolation in both hemispheres, aerosol and
greenhouse gas emission, and land use change forcings over the period 850-2005 CE (Landrum et
al., 2013; Sueyoshi et al., 2013; Yukimoto et al., 2019).

We omitted the glacier on Keele Peak, as its RGI outline includes several cirque glaciers separated from the main glacier, which causes OGGM to produce a problematic flowline that crosses several flow divides. We set the mass balance gradient for each glacier to 5.2 mm w.e. m$^{-1}$ based on the mass balance gradient for Bologna Glacier in Nahanni National Park Reserve for the 2014-2015 CE balance year (Ednie and Demuth, 2019). For each GCM, we ran 300-500 simulations incrementally perturbing the temperature bias (Tbias) and unitless precipitation factor (Pbias) to determine which combination of temperature and precipitation bias produces a modeled glacier length time series that best fits our glacier chronology. Tbias values ranged from -5 to +2 °C and Pbias between 1.0 and 4.0. Initial testing prior to running the larger simulations showed that Tbias and Pbias values beyond the above range produced glacier extents that far exceeded the late Holocene maximum extent of the glacier or made them disappear entirely. When the glacier flowline exceeded 80 grid points beyond the modern glacier extent, the simulation was discarded. For each simulation, we calculated the summed root mean squared error (RMSE) of modeled glacier length versus the moraine and remotely sensed glacier length at multiple timesteps. The combination of Tbias and Pbias that produced the lowest RMSE was selected as the "optimized" set of parameters for each glacier and GCM. The exact values of Tbias and Pbias are not meant to convey specific information about past climate. These values allow for regional tuning of the OGGM model to better fit the reconstructed and observed glacier response.

300

Finally, we averaged the set of Tbias and Pbias from each glacier that produced the lowest RMSE for each GCM and applied those corrections before running simulations of the past millennium for all (1,235) glaciers in the eastern YT/NWT, forced by each "calibrated" GCM. The past millennium climate is of interest as it covers the onset and termination of Little Ice Age cooling. We start all past millennium runs at 1000 CE. We then compared the modeled glacier volume change over the past millennium to our chronology as well as what is already known about late Holocene glacier change in this region to evaluate if the modeling results were reasonable.

### 3.6.4 Future glacier simulations

To predict the fate of glaciers in this region, we use OGGM to project 21st-century glacier change for all 1235 glaciers in the eastern Yukon and Northwest Territories, forced by four different

CCSM4 projection runs under different representative concentration pathways (RCPs). We use the
default model parameters of OGGM v1.5.3 and rely on OGGM's pre-processed glacier directories,
which already contain glacier geometry and climate data.

The historical climate data is CRU TS version 4.04 (Harris et al., 2020). We then download the
CMIP5 (CCSM4) climate model output from four different RCP's and run OGGM's bias
correction against the CRU calibration data, which in turn calculates anomalies from the CRU
reference climatology (1961-1990 CE). Finally, we run OGGM for all 1235 glaciers forced by the
calibrated climate scenarios from 2020 to 2100 CE and analyze the projected change in glacier
area and volume.
**4 Results**
**4.1 Glacier chronology**
Glaciers in the Mackenzie and Selwyn mountains deposited moraines fronting cirque and valley
glaciers 0.7 to 2 km beyond their ca. 2020 CE extents. These moraines are typically devoid of
vegetation other than widespread lichen cover. The moraines we sampled are commonly boulder-
rich, with pebble-cobble matrices (SM Data).

Many alpine cirques preserve two nested moraines within tens of meters of each other. We
observed nested moraine crests at Keele Peak, Arrowhead, North Moraine Hill, and Mordor
glaciers. There is also a partially-nested crest preserved at Anderson Glacier. We did not sample
both crests at most locations since our focus was to date the outermost moraines.

Erratic boulders 10-40 m beyond cirque moraines at Anderson and Mordor glaciers date to 10.9-
11.1 ka (Table 1). An erratic sampled ~250 m beyond the late Holocene moraine fronting Keele
Peak glacier dates to $11.6 \pm 0.3$ ka. Erratic boulders directly overlaid bedrock and had abundant
lichen cover. We did not observe any obvious signs of boulder surface erosion, such as
grüssification, solution pitting, or enhanced relief of resistant minerals.

In the Nahanni National Park region, the median $^{10}$Be age on moraine boulders is $610 \pm 850$ a (ca.
1405 CE, n = 19). Adjacent to Keele Peak, the median moraine exposure age is $370 \pm 110$ a (ca.
1650 CE, n = 8). Together, the sampled moraines in this study date to $460 \pm 415$ a (ca. 1560 CE).
We sampled both the inner and outer crest of the moraine couplet at Arrowhead and Mordor
glaciers. At Anderson Glacier, the outer moraine dates to $390 \pm 50$ a (1620 CE, n = 3) and the
inner moraine to $150 \pm 24$ a (1860 CE, n = 2). At Mordor Glacier, the outer moraine dates to 1260
$\pm 295$ a (760 CE, n = 3) and the inner moraine dates to $390 \pm 22$ a (1630 CE, n = 2).

There is notable scatter in the exposure ages on many of the sampled moraines (Table 1, Fig. 3).
At Nahanni 01, Butterfly, Anderson, Mordor, and North Moraine Hill glaciers, there is at least one
sample from each moraine that returned ages older than 1 ka. This scatter gives individual moraine
ages large errors, however when we analyze all moraine boulder ages together, there is a distinct
peak in exposure ages between ~800 to 100 a exposure (ca. 1200 to 1900 CE), with the greatest
peak around 480 to 280 a (1540-1740 CE, Fig. 3).

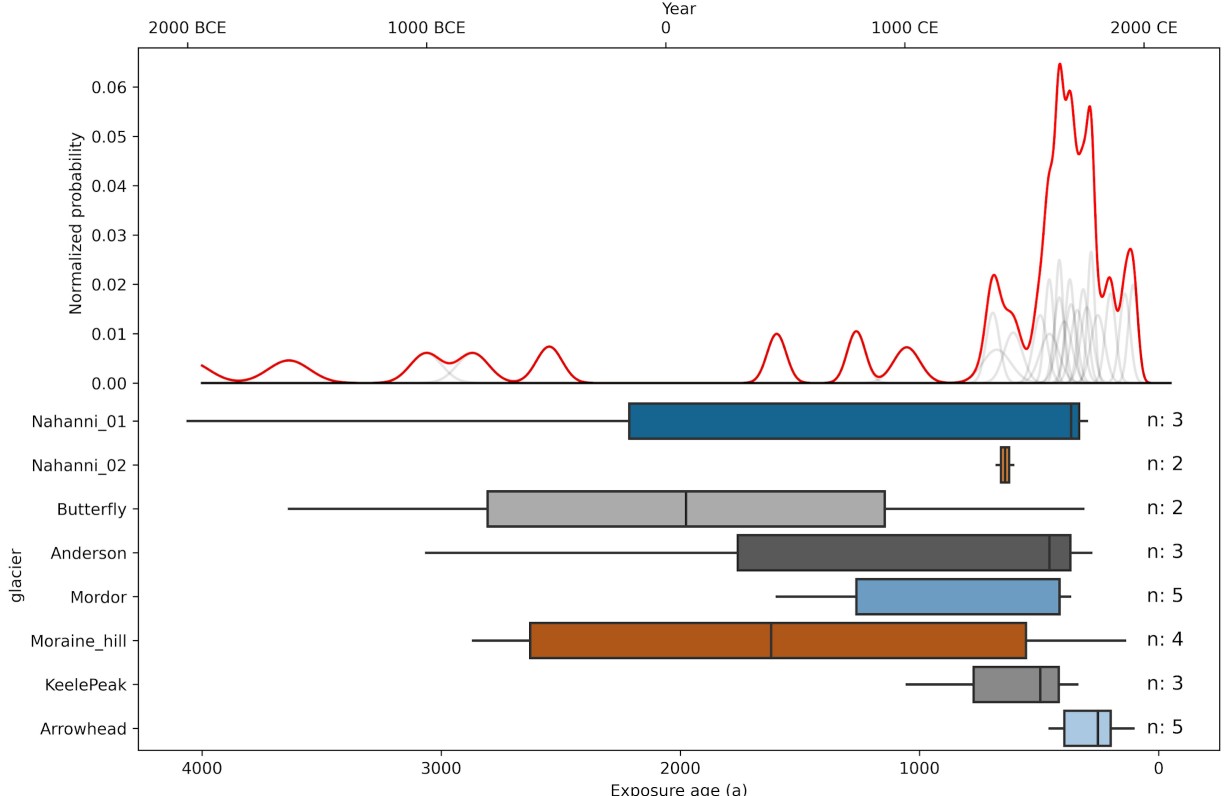


**Figure 3: Box and whisker plots of [10]Be surface exposure ages for each glacier, showing the interquartile range and median**
**age of each moraine surface and the normalized probability density function (red line) for all [10]Be samples and kernel**
**density plot (grey lines) for each individual [10]Be sample.**
**4.2 Climate reconstructions since the late Holocene**
ELA reconstruction using the different methods described above yield a range of estimated
changes in ELA between the LIA and modern time (Fig. 4). We use ELAs from the AAR method
using mapped former and modern glacier extents as the "standard" ELA against which we compare
our other ELA estimates. Any ELA reconstruction method could serve as the "standard"; the AAR
method was selected due to its common usage in glacier reconstructions (Benn et al., 2005; Dahl
and Nesje, 1992; Oien et al., 2022). When comparing ELA change within a single method, "dELA"
is the change in reconstructed ELA between the LIA and modern time using the method in
question. As discussed more below, we assume that precipitation remains constant between the
LIA and modern time for ELA reconstructions using the MELM, THAR, and AAR methods.

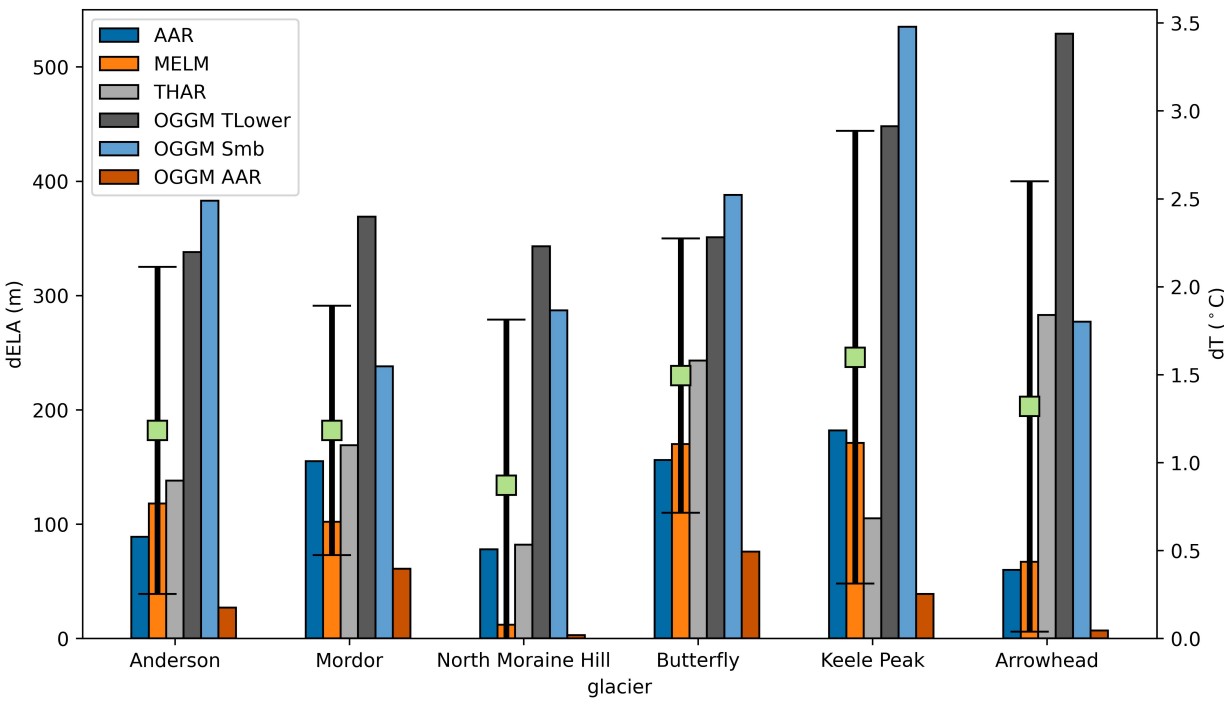


**Figure 4: Changes in ELA and estimated temperature change between the Little Ice Age maximum to modern (ca. 2015)**
**for six glaciers in this study.** Each bar represents a different ELA reconstruction method as described in text. OGGM TLower is
the temperature lowering from ca. 2000 CE climatology required to allow the modeled glacier to reach their late Holocene
maximum extent. OGGM Smb is the change in ELA where the modeled surface mass balance on the glacier equals zero between
the late Holocene maximum and ca. 2000 CE. OGGM AAR is the difference in AAR-derived ELA from the modeled glacier extent
at the late Holocene maximum and ca. 2000 CE. Green squares with capped error bars are the mean and 1-sigma standard deviation
for all ELA reconstruction methods for each glacier.

The modern ELA derived from the AAR method is +12 m to +171 m (average 107 m) higher than
the LIA ELA using the maximum elevation of lateral moraines method, corresponding to a +0.1
to +1.1 °C (average 0.9 °C) increase in temperature (Fig. 4). Using the THAR method, the dELAs
range from +47 m to +240 m (average 138 m), corresponding to a dT of +0.3 to +1.6 °C (average
0.9 °C) since the LIA.

ELAs reconstructed from LIA and modern glacier extent mapping, assuming an AAR of 0.6,
indicate a rise in ELA since the LIA of +60 to +182 m, corresponding to a +0.4 to +1.2 °C (average
0.8 °C) increase in annual average temperature (Fig. 4).

Using OGGM, we include three estimates of ELA change. Non-transient simulations on glaciers
in the Nahanni National Park region using OGGM require +2.3 °C of warming, relative to the 30-
yr average climate centered around 2000 CE, to retreat from their LIA extents to modern positions.
Keele Peak and Arrowhead glaciers require nearly +3.2 °C average warming since the LIA relative
to their modern temperature (Fig. 4). This warming is equivalent to a dELA since the LIA of +354
m in Nahanni National Park and +492 m in the Keele Peak area.

Applying the AAR method, but with OGGM-derived glacier hypsometries at the LIA and modern
time, indicates much less warming since the LIA, with rises in ELAs between +7 m and +76 m,
corresponding to a rise in temperature of <0.1 to 0.5 °C. We interpret this minimal change in ELA
to be the result of glacier surface thickening in the OGGM model when the glacier expands to LIA
extents, which reduces the apparent ELA change as the lower portion of the modeled glacier
surface thickens (SM Fig. 5 & 6).

The third variation of ELA reconstruction using OGGM estimates the modern ELA not from
modeled glacier hypsometry, but rather the elevation at which the modeled surface mass balance
on the glacier is equal to zero. In a warming climate, this estimate of glacier ELA is expected to
be higher than the AAR-derived ELA, as a glacier undergoing rapid retreat has a morphometry
that lags behind the climate signal. Changes in ELA using the modern mass balance-derived ELA
and the AAR-derived LIA ELA range from +277 m to +535 m. Estimated temperature change
indicates a rise in temperature since the LIA of +1.6-3.5 °C.

Using the equation of Ohmura et al. (2018) and temperature change estimates from our AAR-
derived ELAs, we estimate that compared to modern values, there was -117 to -339 mm w.e. yr$^{-1}$,
or 5-15% (average 10%), less precipitation at the ELA of our study glaciers during the LIA (SM
Table 2).
**4.3 Past millennium glacier change**
Estimates of glacier evolution in the YT and NWT over the past millennium vary among the four
GCMs (Fig. 5). The MPI simulation shows steady glacier volume until 1600 CE, while MRI,
MIROC, and CCSM4 indicate a reduction in glacier volume until ca. 1250 CE, afterwards CCSM4
and MRI (and to a lesser degree MPI) show an increase in glacier volume until ca. 1400 CE before
a period of stable ice volume until ca. 1600 CE. MRI, MPI and CCSM4 all indicate glacier
expansion ca. 1600 CE, with MPI reaching a maximum ice volume of 38.1 km$^3$ at 1765 CE and
CCSM4 producing a maximum ice volume of 34.7 km$^3$ at 1855 CE (Fig. 5). MRI appears to largely
miss 20th century glacier retreat and continues to show glacier expansion until 1980 CE, followed
by volume loss. Glacier volume simulated by MIROC decreases through the past millennium, in
contrast to the other GCM simulations. Projections of future glacier loss (below) using CCSM4
climate simulations begin with an initial regional ice volume of 18.1 km$^3$ in 2019 CE. Compared
to the maximum modeled ice volume in the CCSM4 past millennium simulations, this represents
a 48% loss in ice volume since ca. 1850 CE.

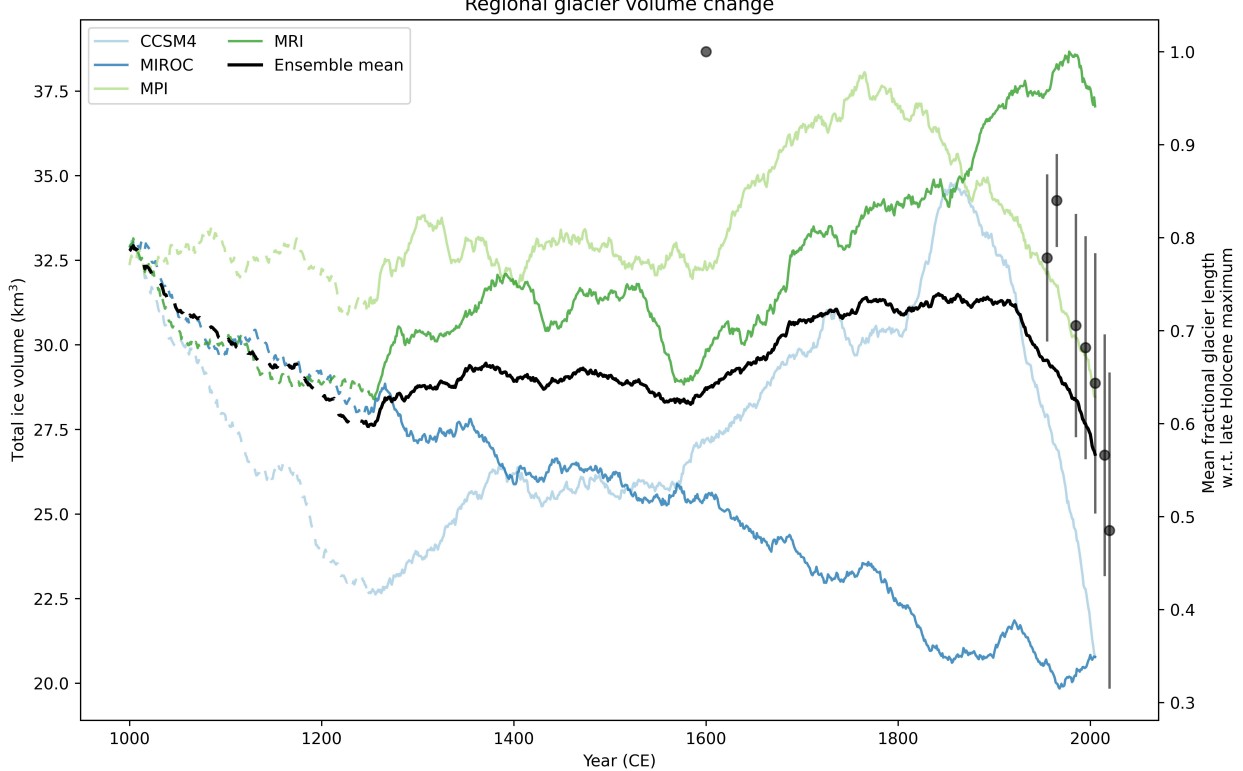

**Figure 5: Modeled ice volume change for all glaciers in the eastern YT and NWT produced by OGGM using four different GCMs.** Dashed lines from 1000 CE to 1250 CE are used to indicate spin up duration of the model. Dots and vertical lines respectively denote average and standard deviation (1-sigma) of normalized mean glacier length binned by decade.

### 4.4 21st Century glacier projections

Under all CCSM4 21st century emissions scenarios, glacier volume in the eastern YT and NWT significantly declines throughout this century (Fig. 6). Glacier volume is projected to decrease by 85% under RCP2.6 and 97% under RCP8.5, compared to 2019 CE values. The greatest rate of ice loss is projected to be between present day and ca. 2040 CE, then the rate of volume decline slowly decreases through to the end of the century.

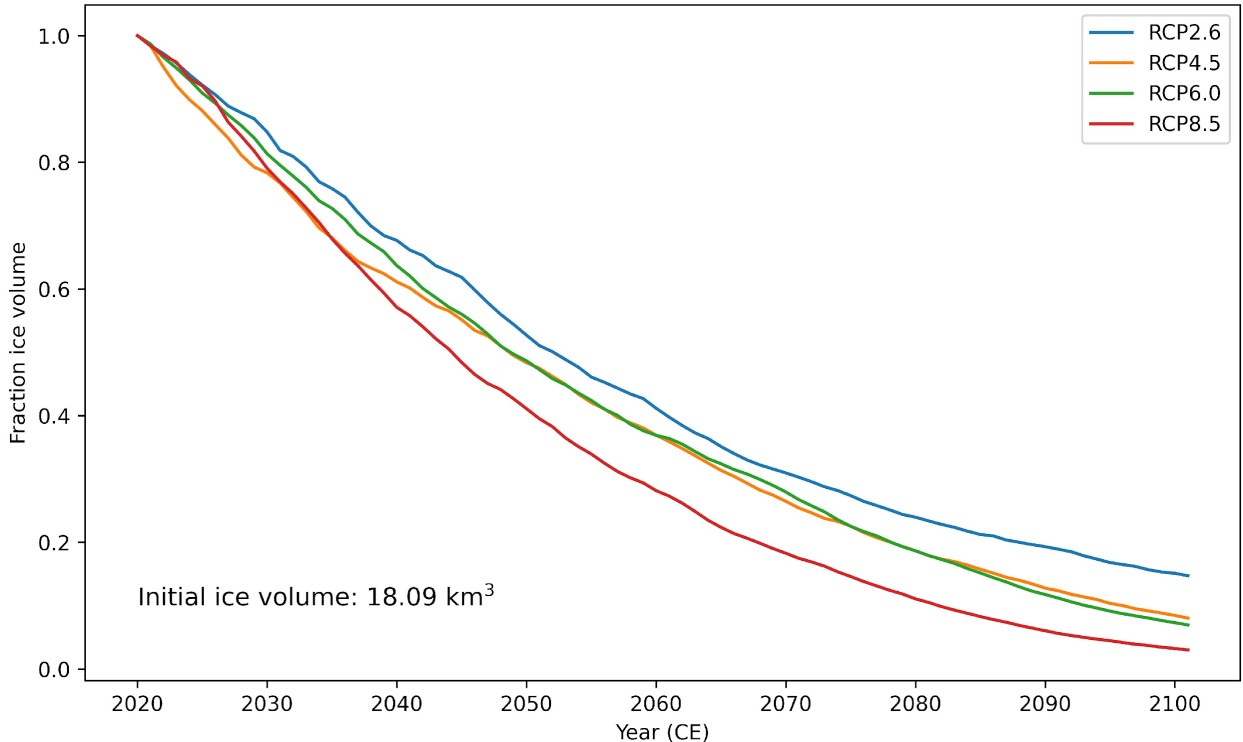

**Figure 6: Fractional glacier volume change until 2100 CE under various representative concentration pathways (RCPs) for all glaciers in the eastern YT and NWT.**

## 5 Discussion

### 5.1 Holocene glacier fluctuations

Early Holocene erratic boulders just beyond moraines dating to the last millennium, as well as a lack of moraines down valley of the latest Holocene moraines, implies that since ca. 11 ka, glaciers in this region were no more extensive than during the latest Holocene. These results accord with records from southern Alaska and western Canada (Menounos et al., 2009; Mood and Smith, 2015; Barclay et al., 2009) that show most alpine glaciers within these regions reached their greatest Holocene positions during the last several hundred years. We interpret the erratic boulders of latest Pleistocene age to record local deglaciation associated with the termination of the Younger Dryas cold interval (Menounos et al., 2017; Seguinot et al., 2016; Braumann et al., 2022). Similar erratic boulders that lie beyond late Holocene cirque moraines were dated by Menounos et al. (2017) and were also interpreted to record local deglaciation. The erratic boulders sampled in the present study were not part of a moraine, so their ages are interpreted to reflect deglaciation at those sites; the absence of an associated moraine precludes us from drawing conclusions about the size of the up

valley glaciers. The most parsimonious explanation for coeval ages of erratic boulders and end
moraines is the complex decay of the Cordilleran Ice Sheet; some cirques were still covered by the
ice sheet while others were ice free prior to the Younger Dryas and so were able to form an end
moraine (Menounos et al., 2017).

Our moraine chronology generally accords with the limited previous work in this region. Moraine
ages from this study suggest glaciers reached their LIA maximum closer to 1560 CE, with a
possible readvance or standstill in the mid-1800's. Tomkins et al. (2008) used varve and tree ring
records near Tungsten, YT to infer periods of glacier growth around the late 1300s to 1450 CE,
1600 to 1670 CE, 1730 to 1778 CE, and an apparent Little Ice Age maximum 1778-1892 CE. Dyke
(1990) completed an extensive lichenometric survey of rock glaciers and late Holocene moraines
directly west and south of Tungsten, dating most late Holocene moraines to within the past 400
years. Our moraine chronology is in general agreement with the lichenometric ages of Dyke (1990)
and suggests an earlier Little Ice Age maximum than interpreted by Tomkins et al. (2008). The
significant scatter in our $^{10}$Be moraine dataset complicates our interpretations of decadal-to-
century scale glacier fluctuations, however.

Several scenarios could yield moraine exposure ages that are either older or younger than the
true depositional age of the moraine. Inherited nuclides from episodes of previous exposure
would result in exposure ages older than the true depositional age. One source of inherited
nuclides could be from rockfall followed by supraglacial transport before deposition on the
moraine. It is also possible that there was insufficient resetting of the $^{10}$Be inventory in the local
bedrock during the Last Glacial Maximum (LGM) as these sites sit at the periphery of the LGM
extent of the Cordilleran Ice Sheet. A third possibility is that the inclusion of old outliers reflects
the incorporation of previously exposed boulders within the glacier forefield. A review of
Holocene glacier fluctuations in western Canada revealed a progressive expansion of ice that
culminated with climatic advances during the Little Ice Age (Menounos et al., 2009). Given what
is known about Holocene glacier activity, the most likely explanation for our pre Little Ice Age
boulder ages is that these boulders contain inherited nuclides from previous moraine building
events and were subsequently reincorporated into the late Holocene moraines during the
advances of the Little Ice Age.

A final possibility to explain the scatter in our moraine ages is that many boulder ages are too
young. Mass shielding by previous burial within a moraine followed by exhumation of a sampled
boulder, or from snow cover, would reduce the nuclide production rate and result in erroneously
young exposure ages. Exhumation and post-depositional movement would be more likely if our
moraines were originally ice cored (Crump et al., 2017).

Snow cover results in younger apparent ages on moraine boulders, however unrealistic quantities
of snow cover are required to meaningfully impact the exposure age of our moraines. One meter
of 0.25 g cm$^{-3}$ snow on the surface our boulders for four months of the year would decrease the
calculated age by 15-27% (SM Table 4). This decrease in age does not significantly impact our
interpretations, as the moraines would still predominately date to the Little Ice Age.

The timing of glacier fluctuations in the eastern Yukon and Northwest Territories agrees with
records of late Holocene glacier advance in Europe (Braumann et al., 2020, 2021; Ivy-Ochs et al.,
2009). Though Europe has different climate forcings than western North America, the similar
timing of late Holocene glacier response suggests that lower temperatures associated with
decreasing summer insolation in the Northern Hemisphere played an important role in the timing
of glacier advance in the late Holocene in both regions.
**5.2 ELA and climate reconstruction**
In this study, we reconstructed and estimated past and present glacier ELAs through several
methods, inline with recommendations by Benn et al. (2005) that multiple ELA reconstruction
methods be used to provide a more robust estimation of past ELAs and uncertainty with each
reconstruction method. An important limitation to the AAR and THAR methods is that they do
not account for modern glaciers being out of equilibrium with modern climate. If the modern ELA
is not accurately known and the glacier is retreating or advancing in response to climate
perturbations, then comparisons in ELA change between modern and other time periods will
under- or over-estimate ELA departures (Porter, 2001). Additionally, the assumption that a
glacier's ELA only fluctuates due to changes in temperature is an oversimplification (Ohmura et
al., 1992). Increased (decreased) precipitation will lead to a higher (lower) mass balance and may
obscure the impact of temperature change on glacier response (i.e. Shea et al., 2004).

Anderson et al. (2011) presents lacustrine $\delta^{18}O$ records from the central Yukon that suggest a wet,
early Little Ice Age, then dry conditions until modern day, in response to the changing position
and strength of the Aleutian Low. If glaciers in the Mackenzie and Selwyn Mountains received
greater snowfall during the LIA, then less cooling would be needed to grow glaciers to their LIA
extents. Tomkins et al. (2008) developed a July mean temperature reconstruction from tree rings
and varved lake sediments close to Tungsten, near the northern end of Nahanni National Park
Reserve. Their amalgamated temperature reconstruction demonstrates the differing signals of
varved lacustrine sediment and tree ring records but does suggest cooler temperatures in the early
1800's, a warm interval at the end of the 1800's to early 1900's, followed by cooling until at least
the 1940's before warmer than average July temperatures until modern time.

Our non-transient experiment using OGGM provides another estimate for temperature change
since the LIA, though it still ignores the effect of precipitation variability. By determining the
temperature lowering from the present climate needed to grow a modeled glacier to LIA extents,
we remove the likely erroneous estimation of the modern glacier ELA based on current glacier
hypsometry and more directly compare modern temperatures with the inferred temperature during
the LIA maximum, when the glacier was in equilibrium with climate. Both the non-transient
("OGGM TLower" in Fig. 4) and surface mass balance ("OGGM Smb" in Fig. 4) incorporate
modern climatology and as a result indicate generally greater temperature change since the LIA
compared to glacier geometry-based reconstruction methods. A bedrock borehole temperature
reconstruction (62.47° N, 129.22° W) between Nahanni National Park and Keele Peak indicates
around +3 °C of surface warming since 1500 CE (Huang et al., 2000), consistent with our
temperature change estimates comparing past ELAs to modern climatology. A similar study design
as presented in this manuscript would be improved by selecting a site with a multi-year *in situ*
mass balance record to compare the modelled modern ELA estimate with the ELA derived from
*in situ* measurements.

OGGM is built to perform best at regional to global scales and may produce problematic results at
the scale of individual glaciers (Maussion et al., 2019). Differences between the year of DEM
acquisition and RGI glacier extent, erroneous glacier margins, and lack of nearby mass balance
calibration information can all have significant impacts on the evolution of individual modeled
glaciers. To help give confidence that the modeling results from OGGM were producing
reasonable glacier evolution, we ran a simple flowline glacier model modified from Jarosch et al.
(2013), which was able to grow glaciers to similar extents as OGGM (SM Fig. 2). The similar
glacier evolution between the two models indicates that modeled glacier response is the result of
climate inputs, rather than unique properties of each model.

As mentioned above, regular mass balance data from *in situ* mass balance measurements or remote
sensing on glaciers in remote areas will help improve the performance and validation of global
glacier models like OGGM (Eis et al., 2021). A similar study design as is presented in this paper
may be successfully implemented in areas with robust glacier chronologies from the late Holocene
to present from many more glaciers than are included in our study. Well-constrained glacier
chronologies would serve to extend the calibration or validation dataset for large scale glacier
modeling efforts (i.e. Rounce et al., 2023).
**5.3 GCM evaluation**
Of the four different CMIP5 GCM simulations tested, glacier model runs forced by CCSM4 and
MPI yield glacier fluctuations that best match our general understanding of latest Holocene glacier
expansion and glacier retreat over the past millennium (Menounos et al., 2009; Luckman, 2000;
Figure 5). We consider the results from MRI to be unreasonable due to the continued ice expansion
through most of the 20th century, and similarly discount the results from MIROC due to the
modeled steady glacier volume decline over the entire past millennium.

Our [10]Be chronology suggests glacier advance and moraine formation earlier than what the
modeling results show. At Arrowhead Glacier, the outer and inner moraine [10]Be ages (1620 and
1860 CE, respectively) are comparable with the modeled glacier evolution under the CCSM4
climate, however. MRI suggests a period of glacier retreat shortly before 1600 CE, which is
consistent with our moraine chronology, however MRI, CCSM4, and MPI all suggest further ice
expansion which would have overridden previously deposited moraines. If the exposure age of a
moraine is interpreted to more closely record the onset of glacier retreat, rather than advance, then
our moraine chronology further indicates that glaciers reached their LIA maximum extents prior
to when OGGM suggests.

The four GCMs used in our study simulate varied temperature and precipitation time series over
the past millennium, which results in differing modeled glacier responses (SM Fig. 8-11). Modeled
glaciers forced by CCSM4 and MPI reach late Holocene maxima between 1765 and 1860 CE,
coincident with other late Holocene glacier records (Menounos et al., 2009; Barclay et al., 2009;
Mood and Smith, 2015). Our moraine and remote sensing record allowed for four GCM's to be
calibrated for a small selection of glaciers in the region prior to being run for all 1235 glaciers.
Without a well-dated moraine chronology, we would be unable to assess how to model performs
beyond the remote sensing record.

Further research is needed to evaluate why the existing GCM simulations fail to grow glaciers at
the same time as our moraine chronology suggests in northwestern Canada. The moraine record
offers an important method of validating glacier models beyond the remote sensing record,
however moraine chronologies must be tightly constrained in order to confidently evaluate model
results. Additional cosmogenic surface exposure dating in this region, especially in areas where
there is an unambiguous lack of post-depositional movement may help to produce moraine
chronologies with less scatter. Measuring multiple nuclides on moraine boulders (such as using
paired $^{14}$C/$^{10}$Be) would allow potential inheritance to be investigated (i.e. Goehring et al., 2022).
Finally, as mentioned above, consistent mass balance records from glaciers in this region would
help to better constrain the influence of local climate on glacier response in the Mackenzie and
Selwyn Mountains (Pelto et al., 2019; Ednie and Demuth, 2019).
**5.4 Future response of glaciers to climate change**
The Mackenzie and Selwyn mountains are almost certain to experience profound glacier mass loss
throughout the 21st century. The estimated magnitude of ice volume decline agrees with modeling
results by Clarke et al. (2015) who estimate a 70-95% reduction in glacier volume in the Canadian
Rocky Mountains by 2100 CE. Additionally, recent work by Rounce et al. (2023) estimates 93-

100% deglaciation in the Mackenzie and Selwyn Mountains by 2100 CE, depending on the magnitude of global temperature change. Under SSP3.7 and SSP5.85, this region is predicted to be fully deglaciated by 2080 CE (Rounce et al., 2023). By 2019 CE, approximately half of the ice volume was lost in the Mackenzie and Selwyn Mountains in the CCSM4 run compared to the glacier maximum in 1860 CE (Fig. 5). The loss of glaciers in this region will cause greater fluctuations in streamflow and temperature that may have negative impacts on thermally stressed species, including fish that are important food sources for local communities (Babaluk et al., 2015; Clason et al., 2023; Moore et al., 2009).

**6 Conclusions**

Based on geomorphic mapping, surface exposure ages, and numerical modeling, the following conclusions can be drawn from our study. (1) The probability distribution of $^{10}$Be ages suggests that most glaciers in eastern YT and NWT reached their greatest Holocene extents during  the latter half of the Little Ice Age [1600-1850 CE]; (2) The uncertainty ascribed to some moraines is high, given the presence of some boulders that yielded $^{10}$Be ages that predate the Little Ice Age, and future work utilizing multi-nuclide approaches would allow this scatter to be further investigated; (3) We find no evidence of glaciers extending beyond LIA limits since at least 10.9-11.6 ka, in accord with most other Holocene glacier records in the Northern Hemisphere; (4) Our ELA reconstructions suggest warming of 0.2-2.3 °C since the LIA, with morphology-based ELA reconstructions likely underestimating the modern ELA of glaciers undergoing retreat; and (5) Projections of future glacier change estimate a further 85-97% loss of glacier volume in the Mackenzie and Selwyn mountains by 2100 CE, in agreement with recent global modeling efforts.

Glacier chronologies from late Holocene glacier fluctuations can provide important sources of validation of GCM simulations beyond the instrumental record, especially given the variety between individual GCM simulations of past climate. Nearby *in situ* mass balance records and well-constrained late Holocene glacier chronologies are needed to help validate past millennium GCM simulations and highlight important feedbacks between the arctic and the global climate system. Modern tropospheric warming will continue to dramatically reduce glacier volume in this region, with significant impacts to the local ecosystem that relies on glacier-fed rivers and streams through the summer months.


*Author Contributions.* Following the CRediT Authorship Guidelines, AH contributed to all 14 authorship components except resources and supervision. BM was involved in all authorship components. BG contributed to formal analysis, investigation, resources, supervision, validation, and review/editing. GO was involved in conceptualization, investigation, supervision, and review/editing. BP contributed to data curation, methodology, and software. CD was involved in investigation, visualization, and review/editing. JS was involved in conceptualization, funding acquisition, investigation, and review/editing.









*Competing Interests.*


The authors declare that they have no conflict of interest.



*Acknowledgements.*


Funding for this study was provided by a NSERC Northern Supplement and Discovery grant to BM, and a GSA Quaternary Geology and Geomorphology Division Arthur D. Howard Research Award to AH. Additional travel support was provided to AH by the University of Northern British Columbia. The Geological Survey of Canada shared helicopter access in the Nahanni National Park Reserve (NNPR) and graciously allowed us use of their concrete saw. The friendly staff at the Whitehorse Airphoto Library provided invaluable assistance with field site reconnaissance. We are grateful to the Dehcho, Denendeh, and Nahanni Butte First Nations for access to complete our study on their traditional territories. Rebecca Lerch assisted in field work in NNPR. Expert flying by Alpine Aviation provided floatplane access to remote sites around Keele Peak and in NNPR.












*Code and data availability.*


All data described in this paper that have not already been published elsewhere are included within the main text and/or supplementary materials. Code used for glacier modelling has been sourced from OGGM.org or from Jarosch et al. (2013). In the event of paper acceptance and publication, the code will be posted on a publicly available repository under an open-source license.

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
