# Peer review of "Late Holocene glacier and climate fluctuations in the Mackenzie and Selwyn Mountain Ranges, Northwest Canada"

_The Cryosphere, 2023_

## Author Comment (AC1)

Reviewer Comment 1:
Christopher Halsted
**General Comments**

This manuscript represents a valuable contribution to our understanding of Holocene glacier chronologies and regional paleoclimate fluctuations in northwestern Canada, particularly given the relative lack of empirical data from this remote region. The glaciers and moraines targeted by the authors are well-suited for the study objectives, an impressive feat given that surveying was done through satellite and aerial imagery. The methods are generally appropriate, although I have some critiques about how the [10]Be exposure ages were statistically interpreted (see following sections). I am not as familiar with ELA reconstructions or climate modeling as I am with exposure dating, but the methods, assumptions, and applications seem reasonable as conducted here. The authors do a good job of comparing their interpretations of Holocene glacier chronologies to other nearby glacier and paleoclimate records, providing a nice synthesis of climate change in the past millennium in northwestern North America.

The foundation of this manuscript is solid, but there is some work that needs to be done organizationally and in terms of data analysis before I can recommend it for publication. I outline my specific comments below. I hope that the authors find these comments to be constructive and helpful, rather than onerous.

**Thank you for taking the time to review our manuscript and for your helpful comments and insights. Below, we discuss and address your comments and incorporate most of your suggestions into our revised manuscript. We reply to individual comments in bold font.**

*Specific Comments*

Aside from smaller technical comments, I have two more substantial and specific critiques for the authors to consider.

First, this manuscript does not have a background section, but I believe that it would benefit from one. As written, a lot of background information is sprinkled between the methods, results, and discussion sections, such that the methods section is very long (6 pages) and some much-needed background about the methods being used is introduced *after* the results have been presented. The existing "Study Area" section, which currently consists of a single paragraph, could also be wrapped into the background. You might also consider adding some field or site photos to this background section, especially because your field area looks stunning (perhaps your SM Figure 7?). I have noted in the "Technical Comments" section the specific lines that I identify as being more background than methods and could be re-located to a background section. Additionally, there is currently limited background about [10]Be exposure dating, although it is a key component of this study. Consider expanding the background information about exposure dating, including the issue of inherited nuclides causing age scatter that is so prevalent in glacial moraine chronologies (see Balco, 2020, in *Annual Review of Earth and Planetary Sciences* for a great overview). As is, inheritance is only mentioned once at the very end of the discussion, but I believe that it plays a substantial role in some of the older exposure ages observed on moraines in this study.

**Thank you for this point. While we recognize the need to provide the reader with important background information, we chose not to include a separate Background section to maintain the conventional structure of manuscripts submitted to the Cryosphere. In the original draft, however, we acknowledge we presented significant "background" only within Results and Discussion. We moved our background information to portions of our Introduction, Study Area, and Methods as appropriate to provide that key information to the reader prior to the Results and Discussion sections.**

On that note, I have some critiques about how moraine abandonment dates were estimated from $^{10}$Be ages. The significant variation in ages on several moraines suggests some source of geologic scatter, rather than just being due to analytical uncertainties, but the potential causes of this scatter are not considered. Rather, all ages are used to estimate moraine ages, causing 1) considerable disagreement between some moraine ages and 2) some very large age uncertainties, especially for moraines with several older exposure ages (e.g., Butterfly, Mordor outer, and North Moraine Hill glaciers). In my opinion, there are two plausible explanations for the observed exposure age scatter, and they bear consideration at some point in the manuscript. First, boulders with older exposure ages (~1 to 4 ka) may contain varying amounts of $^{10}$Be inheritance. If your sampled glaciers were indeed less extensive for the majority of the Holocene than during the LIA, these boulders may have been exposed on the proglacial landscape for thousands of years, accumulating $^{10}$Be. During the LIA, the boulders would have been re-worked onto the moraines as glaciers advanced, but they may not have been entirely stripped of their Holocene $^{10}$Be. If this history is correct, the scatter observed among these older ages likely reflects both re-orientation and varying degrees of glacial erosion experienced by these boulders during LIA re-working. Another potential mechanism to explain the geologic scatter is that the younger ages reflect post-deposition processes that result in partial shielding or disturbance of moraine boulders, thus causing their ages to be younger than the true moraine abandonment date. In this case, the older moraine boulder ages would more accurately reflect the dates of moraine abandonment, and the young ages are 'red herrings'. In my opinion, the first explanation (older ages have inherited $^{10}$Be) is far more plausible, especially given the tight distribution observed in your younger ages across moraines and the contrastingly large distribution of older exposure ages (as demonstrated well in figure 3).

I say the above not to be overly critical, but because I genuinely believe that you have a valuable dataset here and that significant results are being overlooked because of the analyses used. If I may offer a suggestion, I'd recommend labelling the older ages as outliers containing $^{10}$Be inheritance and estimating moraine ages using the mean and standard error of the younger ages. Such an approach seems warranted when looking at exposure ages from all sampled moraines together. If we assume that all of these moraines correspond to approximately the same paleoclimate event, and are thus of similar age, then the distribution of exposure ages shown in Figure 3 clearly demonstrates that the young ages are tightly clustered while older ages exhibit quite a lot of variance (likely due to varying levels of $^{10}$Be inheritance in sampled boulder surfaces). I think that by using just the young exposure ages, you will get much tighter and more consistent age estimates of moraines, and the overall age estimate of the moraine population will likely become younger as you remove the older samples. You already do this to an extent from lines 285 to 287, where you identify the peak of exposure ages, but I think you can use this peak as evidence to get more accurate moraine ages by discarding old ages.

**Thank you for your thorough explanation of your suggestion. We fully agree that the two hypotheses presented above are logical and feel that hypothesis (1) is the most likely scenario that accounts for some of our oldest ages. We now address those hypotheses in the Discussion section of the paper, but would prefer to maintain our convention on reporting the $^{10}$Be ages for a given moraine using non-parametric methods (e.g. median and interquartile range). The issue we wish to avoid is to arbitrarily remove outliers since to do this objectively requires us to assume an underlying distribution for the moraine boulders (these arguments were brought up by Menounos et al., 2017 and Darvill et al., 2022). Given that our dataset is likely influenced by inheritance or the impact of exhumation, snow cover, or erosion, we are uncertain whether these boulders would yield a normal distribution in ages for a given moraine. While our conservative reporting of ages yields high scatter for a given moraine, the joint probability distribution of ages yields a notable peak for the likely abandonment of the moraines (i.e. moraine stabilization). It is our belief that, with sufficient sites, multiple peaks that appear as outliers for a given moraine, would yield probable ages for earlier advances within a given region. We now bring these points up in the Discussion section of the revised paper.**

*Technical Corrections*

Line 60: The wording of this line is a bit confusing, maybe re-write as "…reached their greatest Holocene positions around 1600-1850 CE, at the culmination of the Little Ice Age (LIA, ~1300-1850 CE)…"

**Changed as suggested.**

Lines 62-64: Consider moving the last line to the beginning of the paragraph, it reads like a topic sentence (which is missing from this paragraph anyways).

**Good suggestion. Paragraph restructured.**

Lines 65-69: The use of a numbering system for only the first two objectives is somewhat confusing. Consider either numbering your third and fourth objectives or get rid of the numbers for the first and second objectives.

**Numbering of the first two objectives removed.**

Lines 79-82: The climatological information as it is presented here does not feel strictly relevant to your study. It becomes relevant later when you discuss the paleoclimate implications of your ELA reconstructions, but that connection is not clear as written. Consider either adding a few lines explaining its relevance to Holocene glacier chronologies, or remove this information here and bring it up at the relevant point in your discussion.

**A valid criticism. We removed this information from the Study Area section and now include this as part of the Results section with additional discussion in the Discussion section.**

Line 84: Remove "To summarize our methods"

**Removed.**

Line 89: Consider adding something like "…and infer changes in temperature and precipitation *from estimated ELA changes*" to clarify *how* you are inferring the paleoclimate changes.

**Added.**

Lines 105 – 107: The wording of these sentences is confusing, because you introduce the glaciers by name and then state that most have no formal name. Could you re-word so that it is clearer that those are your informal names for the glaciers?

**Clarification added.**

Line 105: Should this citation be for SM Table 1?

**Great catch, yes, the numbers on the first few SM Tables/Figures were incorrect and are now corrected.**

Line 125: SM Figure 3 does not seem like the right figure to be citing here (it is a climate model temp and precip bias calibration).

**We apologize for this error in initial submission and now refer to SM data with field photos from each sampled boulder.**

Line 127: Add a reference to support your statement about moraine boulders (I recommend Heyman et al., 2016, *Quaternary Geochronology*).

**Thank you for this. We added the Heyman et al. (2016) reference.**

Line 136: Consider changing this sentence to "We processed samples collected in 2014 at the Lamont-Doherty…" As is, it sounds like it was LDEO itself that was doing the sample processing, rather than you.

**That is correct. We sent the samples from 2014 to LDEO where the staff processed samples, whereas the lead author performed the laboratory work for the remaining samples at the Tulane University Cosmogenic Nuclide Laboratory.**

Line 139: Consider replacing the Nichols and Goehring reference with Kohl and Nishiizumi (1992). The Nichols and Goehring paper was specifically about complications in quartz isolation for *in situ* $^{14}$C exposure dating, which is not relevant to this study. Kohl and Nishiizumi is the original (and still followed) quartz isolation procedure for $^{10}$Be analysis.

**The quartz prep used in this study did follow the recommendations of Nichols and Goehring, rather than those of Kohl and Nishiizumi, though the methods are quite similar.**

Line 141: Should this citation be SM Table 3?

**Changed.**

Line 144: Add a reference to Table 1 somewhere in this sentence, as Table 1 shows which samples were sent to PRIME vs. LLNL. Also, for consistency, either give the abbreviation for LLNL-CAMS after introducing the laboratory, or don't give the PRIME abbreviation in the text.

**Changed.**

Table 1: Consider rounding exposure ages and uncertainties to the nearest decade. Annual precision is not yet feasible with $^{10}$Be exposure dating. Additionally, the caption repeats information given in the table footnotes (about the exclusion of erratic boulders from moraine ages), consider deleting this information in the caption. Finally, grammar edit for footnote d: "excludes the exposure age of erratics, whose *ages are* listed in italics".

**A useful suggestion. Ages and errors are now rounded to the nearest decade and footnotes have been edited as suggested.**

Lines 154-159: All but the last line of this paragraph feels like background information, it should probably not be part of the methods section. See my comments in the "Specific Comments" section about maybe adding a background section, these lines would fit into such a section to guide the reader through using ELAs to reconstruct past climates. Also consider expanding your explanation of "Each method offers advantages and limitations in reconstructing past ELAs". Some readers (myself included) might not know the systematics of these methods and require a bit more guidance.

**We opted to keep this information within Methods and have moved some information only presented in Discussion to this section of the paper.**

Line 156: Remove apostrophe from "ELA's" here and elsewhere.

**Agreed, changed.**

Lines 165-169, 171, and 183-186: The first sentences of the THAR, AAR, and ELA/precipitation paragraphs also feel like background information rather than explanations of *your* methods.

**Understood, please see our previous comments on treatment of "background" information.**

Line 250: "Finally" is used in successive paragraphs, consider removing it from this sentence.

**Removed.**

Lines 262-279: These two paragraphs also feel like background information that should more appropriately belong in your "Study Area" section, or in a dedicated background section.

**The geomorphic description of the moraines, erratic boulders, and change in extent from their late Holocene positions is new information that has not been previously published. Therefore, we retain this information within Results, rather than Study Area information.**

Lines 276-281: The ages of the moraines are likely to change if you follow the suggestions I provided in the "Specific Comments" section, but as is, the large uncertainties on exposure ages should probably be mirrored in the dates you give in parentheses. For example, "610 ± 850 a (ca. 1405 CE)" should realistically read "610 ± 850 a (ca. 1405 ± 850 CE)" or "(ca. 560 CE – Present)"

**Presentation of ages is kept as originally presented. Discussion of our choice of summary statistics is included above.**

The top panel of Figure 3 is great. However, I wonder if box and whisker plots are the best plot option for your individual moraine exposure ages. Particularly for moraines with 2 or 3 exposure ages, these plots do not provide a good visual to see how your ages are distributed, nor the uncertainties on each age. Consider replacing the box and whisker plots with one-dimensional scatter plots (i.e., there is no vertical axis and ages are plotted as symbols with error bars). I recommended in the "Specific Comments" section that you should use mean exposure ages and standard errors for moraine age estimates, rather than the median age and IQR, so a visual of the IQR will be less relevant anyways.

**Discussed in our response to the treatment of moraine ages, however we have also now included individual boulders ages on Figure 3.**

Line 295: The statement about AAR method being commonly used in glacier reconstructions needs a reference or two to back it up.

**Additional references are now added.**

Figure 4 caption: Consider adding a quick explainer for what "TLower", "Smb", and "AAR" stand for in your OGGM runs. I had to go back and forth to the figure as I was reading to figure out which was which.

**Explainer added to figure caption.**

Line 334: Should this cite be SM Table 2?

**Changed.**

Line 360: Should this read "…a lack of moraines *up valley* of the latest Holocene moraines…"? To me, *down valley* suggests moving farther away from the glacier front, not back towards it.

**What we are saying here is that there are no moraines with greater down-valley (farther from the glacier headwall) extents than the dated late-Holocene moraines. This lack of moraines distal to the late Holocene moraines implies that the glacier in question was no more extensive than its late Holocene position for the entirety of the Holocene. We have left this text unchanged.**

Line 363: The European records feel somewhat out of place here. You are focusing on other glacial chronologies in northwestern North America, which all presumably were subject to similar climatic forcings, but glaciers in the Alps would have been subject to far different climatological influences. However, the similarities are certainly interesting! Consider moving this to later in the discussion, maybe a dedicated few sentences or paragraph about similarities between glacier chronologies in northwestern North America and elsewhere in the world that show the prevalence of the LIA throughout the northern hemisphere.

**Moved to later in section 5.1 to form a separate paragraph.**

Lines 379-387: Much of this paragraph feels like important background information that should have been introduced earlier. This could be moved to a dedicated background section.

**Thank you, now presented earlier in the paper.**

Line 382: Grammar edit – "…limitation to the AAR and THAR *methods* is that *they do* not account…"

**Changed as suggested.**

Line 389: This paragraph feels like it is missing a topic sentence. I'd suggest a line summarizing the key points of your ELA reconstruction and what they imply about climatological changes since the LIA.

**Thank you, we have now added a topic sentence.**

Lines 411-414: The first two sentences of this paragraph read like background material.

**Now covered earlier in the paper.**

Supplementary figures 1, 4, and 7 and table 4 are not referenced in the text.

**Good catch, references to these SM Figures and tables have been added to the main text.**

---

## Author Comment (AC2)

Reviewer Comment 2:

Alia Lesnek

In this manuscript, Hawkins et al. present a new [10]Be chronology of late Holocene moraines in northwest Canada. They also reconstruct past regional climate from modeled glacier ELAs, simulate glacier volume over the past millennium, and forecast future glacier volume over the next century. The topic of this manuscript is well-suited for publication in *The Cryosphere*. In all, there's an impressive amount of work that went into this study. The results and interpretations will be a useful contribution to our knowledge of past and future glacier change in Canada.

That said, the manuscript would benefit from revision before publication. Below, I've provided a few 'major' suggestions for the authors to consider; these are not all that major in the grand scheme of things, but may require substantial revisions to the text and some figures. I've also included a number of minor suggestions, indicated on a line-by-line basis. I hope that my comments offer constructive ways to improve the manuscript.

**Thank you for taking the time to review our manuscript and for your helpful comments and insights. We have discussed and addressed your comments below and incorporate many of your suggestions into our revised manuscript. Please see our responses in bold below.**

**Major comments**

*Interpretation of [10]Be ages*: The pre-LIA exposure ages in your chronology are under-discussed in my opinion. I recognize that the number of exposure ages that you have per moraine is somewhat small and you are probably wary of over-interpreting your data, but if you assume that the late Holocene moraines date to the same event (which it seems you do based on your ELA reconstruction and modeling approaches), then I think you can expand your discussion of the exposure ages in at least two major ways.

First, the fact that you have erratics with consistent 11 ka exposure ages just outside of the late Holocene moraines suggests to me that the moraines were emplaced by a glacier readvance rather than a stillstand. Young et al. (2013), *QSR* have a nice discussion about using exposure age distributions to infer that a readvance created some of the Fjord Stade moraines in western Greenland. The distinction between a readvance and a stillstand to create the late Holocene moraines is an important one for your modeling and paleoclimate interpretations, and I think your dataset does allow you to distinguish between these two scenarios.

**Thank you for your thoughts and discussion on our erratic boulder ages. An important distinction between the moraines in Young et al. (2013) and the erratic boulders sampled in this study is that our erratic boulders were not a part of any moraine system. Similar erratic boulders were dated by Menounos et al. (2017) that lie beyond cirque moraines and their ages were interpreted to reflect the age of local deglaciation. Since no landform was associated with these erratics (or those in the present study) we can only report**

deglaciation rather than whether the cirque glacier was more or less extensive. In addition, Menounos et al. (2017) found that some erratics and end moraines dated to the Younger Dryas termination. The most parsimonious explanation for coeval ages and associated geomorphology was the complex decay of the Cordilleran Ice Sheet (i.e., some cirques were still inundated by the ice sheet whereas others were ice free prior to the Younger Dryas and so could form an end moraine). We now bring up this point in the Discussion section of the manuscript.

Second, given that these glaciers likely readvanced to deposit the late Holocene moraines, it seems reasonable that the boulders with exposure ages between 1-4 ka were reworked and therefore have small, but variable amounts of $^{10}$Be inheritance, and that the younger cluster of exposure ages more closely constrains the timing of moraine abandonment in your study area. That said, it's also possible that the younger exposure ages are too young due to snow shielding or exhumation (e.g., although there isn't a lot of debris on the glacier termini now, I could *potentially* see a scenario where the moraines were ice-cored early in their history and the youngest exposure ages are actually reflecting the timing of moraine stabilization in the region). However, to my eye, the tight clustering of young exposure ages across the six glacial systems combined with the scattered "old tail" of ages seems most easily explained by the inheritance interpretation.

**As noted by Referee #1, we concur with your assessment here. We now expand on this point in the Discussion section of the paper.**

These are just two areas where I think you can beef up your $^{10}$Be interpretations. A more thorough discussion of the above ideas (and other relevant ones that you may come up with during the revision process) as they apply to your study area would support the modeling you do later in the paper and strengthen your overall conclusions about glacier history in NW Canada.

*ELA estimation methods:* In line 407 of the paper and in the conclusion, you recommend that modeling modern glacier ELAs using climate data should be preferred over methods that rely on glacier geometry. I'm not fully versed on all of the latest glacier ELA literature, but this seems like an important outcome/recommendation from this study. Perhaps it's outside the scope of this paper or it's already been done by others, but I wonder how ELAs modeled based on climate data compare to ELAs calculated with *in situ* mass balance measurements. Are there any examples of glacial systems where this has been done before? Or even better, any glaciers in your study area with *in situ* mass balance measurements and corresponding ELAs that you could compare with your modeled ELAs to validate this approach?

**Unfortunately, there are few glaciers within our area that have mass balance records for any significant length of time. The Geological Survey of Canada has periodically completed mass balance campaigns on Bologna Glacier at the northern end of Nahanni National Park Reserve. We are unaware of other papers that directly compare ELA's produced by coupled glacier/climate models compared to geomorphic ELA estimations. However, modelled ELAs have been compared with *in situ* ELA measurements (e.g. Braithwaite and Raper, 2015; Keeler et al. 2021). An important limitation to a flowline glacier model, as used in this study, is the glacier bed geometry strongly influences model behavior. In**

**OGGM, bed surface topography is estimated with a surface inversion model, which may differ significantly from the actual glacier bed geometry. In addition to employing glacier models of greater complexity and high-resolution climate models, utilizing ground penetrating radar surveys to map subglacial topography would improve model performance. Implementing these suggestions are outside the scope of this paper, but may be fruitful avenues for future research.**

*Modeling details:* The modeling exercises you did are a useful contribution, but the description of your model setups, particularly for the transient OGGM experiment, could use more detail. Perhaps I missed these things, but for example, where are you getting the starting values for glacier volume at 1000 CE? Are these coming from moraines or some other source? What ranges of Tbias and Pbias did you use to tune the climate models, and how do these bias values compare to what's known about regional climate over the past 1 kyr? And more generally, why simulate the past millennium rather than some other time interval?

**You bring up valid, useful suggestions here. Tbias and Pbias ranges are now added to the Methods section. These biases are adjusted to minimize the difference between the modelled and "observed" glacier extent for each glacier and do not represent "real" climate information and so are not compared to the known regional climate. We chose to spin up the modeling based on our use of the GCM transient runs available to us (Eis et al., 2021; Huss and Hock, 2015). Since these runs commence at 0850 CE, that provides us with 150 years of 'spin up' for the glaciers. We now describe that in more detail within the revised manuscript and will clarify the spin up duration on Figure 5.**

*Figure 5:* It would be helpful to include the known glacial history on this figure so the reader can see how the modeled glacier changes compare to the geologic constraints. I know this figure is showing ice volume for all 1235 glaciers in the study area, but a second panel showing something like a generalized time-distance diagram normalized to glacier length for the glaciers you studied in detail (incorporating data from the moraines and the satellite imagery) would help readers evaluate the results of your modeling.

**This is a great suggestion, and Figure 5 has been edited to show this.**

**Minor comments**

Line 117: Can you include a supplemental figure showing your late Holocene glacier margins? Drawing paleoglacier margins can be quite challenging in the accumulation zone where there are no moraines, so this would be helpful to see.

**We now include a Supplemental Figure showing the digitized margins of the glaciers used in ELA reconstructions. As noted in the figure caption, not all margins include the accumulation zone, as the primary focus was to measure the distance along the central flowline of the glacier to the where the flowline intersects the glacier terminus at each time step.**

Line 145: Are the exposure ages presented in the main text calculated assuming no surface erosion or snow cover?

**We added clarification that snow cover and erosion is not corrected for in the presented exposure ages. We do present exposure ages with a reasonable snow cover for the region in SM Table 4. We expect erosion rates to be low and snow cover of minor importance, given the relatively young ages of the moraines.**

Line 178: Which glacier extents? Modern and LIA? What year did you use for your "modern" glacier extent (since you seem to have digitized glacier extents for many years from Landsat imagery)?

**Clarification added to specify that the "modern" glacier extents are from imagery between 2017 and 2021 CE for each glacier.**

Line 235: This choice of mass balance gradient seems reasonable for a modern glacier, but I wonder how much of an impact that choice makes on your model output, especially since your glaciers advanced into the LIA?

**This is an important limitation to our method and is now addressed in Discussion. We do not have multi-year mass balance gradient observations from nearby glaciers in this region.**

Line 334: This SM table citation doesn't look like it goes to the correct table.

**We apologize for this error. The first few SM tables were out of order in the original submission, now corrected.**

Line 458: Reading this section and the last line of your conclusions, I'm left wanting a bit more information about the wider implications of glacier loss in this region. What specifically are the potential impacts on ecosystems? Are there people who rely on these glaciers for water, etc.?

**Additional information on the cultural and ecological setting of the watersheds in our study area added to the Study Area section. We now note in Discussion that we cannot constrain the impacts to the ecosystem from glacier loss in this region, but we do mention potential impacts to local fisheries that are important to local First Nation communities.**

Figure 2: This figure would be improved by adding your exposure ages and/or sample IDs so your readers can see how the ages are distributed across the moraines. Are the erratic boulders also included on this figure? It's hard to tell.

**Sample ages are now added to Figure 2 along with inset maps that show greater location detail for individual samples, including erratics.**

Figure 3: Can you also show the individual exposure ages on the kernel density plot below the red summed curve? That will be useful for seeing how many ages contribute to a particular peak.

**A great suggestion. We now include individual samples to the figure to better show distribution and contribution to peaks.**

Figure 4: Define the abbreviations Tlower, Smb, etc. in your caption and in the relevant methods sections. I didn't see these terms defined until the discussion.

**Agreed.**

Table 1: Exposure ages should be rounded to the nearest decade.

**Agreed.**